# Examining the Impact of Regional Development Policy on Industrial Structure Upgrading: Quasi-Experimental Evidence from China

**DOI:** 10.3390/ijerph19095042

**Published:** 2022-04-21

**Authors:** Shengli Dai, Weimin Zhang, Yingying Wang, Ge Wang

**Affiliations:** 1School of Public Administration, Central China Normal University, Wuhan 430079, China; storymaker@163.com (S.D.); zhangweimin@mails.ccnu.edu.cn (W.Z.); wangyingying@mails.ccnu.edu.cn (Y.W.); 2School of Administration and Emergency Management, Jinan University, Guangzhou 510632, China

**Keywords:** Yangtze River Economic Belt, upgrading of the industrial structure, industrial structure advancement, industrial structure rationalization, difference-in-differences

## Abstract

“Guiding Opinions on Relying on the Golden Waterway to Promote the Development of the Yangtze River Economic Belt (YREB)”, the “YREB Development Policy”, is a national Chinese policy aiming to promote industrial structure upgrading and high-quality development in the YREB. To test the effects of this policy, we used 2009–2019 data from 283 cities to examine whether its implementation promoted regional industrial structure upgrading. The YREB Development Policy positively impacted industrial structure advancement but scarcely benefited industrial structure rationalization. Moreover, the impacts indicated a temporary, unsustainable influence on industrial structure advancement, presenting a clear U-shaped development trend. The YREB Development Policy can more significantly improve industrial structure upgrading in cities with large populations. The effects of this policy on industrial structure upgrading in the middle and lower reaches of the YREB are almost five times that in the upper reaches. In addition, the policy more greatly impacts the tertiary industry than the primary and secondary industries, especially in the lower reaches. These findings have policy-making implications, enrich the research regarding the YREB Development Policy impacts on industrial structure upgrading, and provide an empirical reference to improve subsequent policies.

## 1. Introduction

Although brilliant achievements have been made in the economic development of China since the reform and opening up, the extensive economic development mode carries high costs that cannot be ignored [1]. The Chinese government has realized that upgrading the industrial structure is an essential way for the Chinese economy to achieve high-quality development at the present stage [2]. To effectively reverse the high costs associated with the current economic development model, the Chinese government is working extensively to eliminate the negative monetary impacts by upgrading the industrial structure.

To ensure the smooth transformation and upgrading of the industrial structure, the Chinese government has formulated ambitious development plans. For example, during the 12th Five-Year Plan period, the Chinese government clarified the direction with which key industries are to be restructured. At the same time, it also proposed requirements regarding the closing of outdated production facilities and the reduction and lessening of excess capacity [3]. During the 13th Five-Year Plan period, the Chinese government paid more attention to the real changes occurring in relevant factor conditions and adopted market-oriented policies to guide the optimization and upgrading of industry [4]. After drawing on the useful experiences of the past, the Chinese government identified four key issues related to the industrial structure during the 14th Five-Year Plan period, that is, solving overcapacity, cultivating emerging industries, reforming the manufacturing industry and optimizing the service industry [5]. These development practices indicate that policy plannings provide effective guidance for China’s future economic development.

As an important gathering place of China’s economy and population, the Yangtze River Economic Belt (YREB) plays an important strategic role in the overall regional development pattern in China [6]. As early as 2014, the State Council of China issued the “YREB Development Policy” as a national strategy. One of its core tasks is to rely on innovation to promote the upgrading of the industrial structure and high-quality development in the YREB. In 2021, the gross domestic product (GDP) of the YREB reached CNY 45.8 trillion, accounting for approximately 49.7% of China’s total GDP. The proportion of added value of the three industries in the YREB also changed from 9.6:46.4:44.1 in 2014 to 8.4:39.2:52.4 at present. Since the implementation of the YREB Development Policy, the primary and secondary industry proportions in the YREB have decreased by 1.2% and 7.2%, respectively, while the tertiary industry proportion has increased by 8.3%. This phenomenon leads to an important question about whether the YREB Development Policy plays an effective role in promoting the upgrading of the industrial structure in the YREB. As such, an in-depth evaluation of the effectiveness of the YREB Development Policy on industrial structure upgrading in this area is of great practical significance.

The difference-in-differences (DID) method is an important tool for policy evaluation that not only avoids the problem of endogeneity of variables, but also mitigates estimation bias caused by missing variables [7]. As a powerful policy evaluation tool, the DID method can accurately quantify policy effects and produce more realistic test results [8]. Applying the DID method to assess the difference in change before and after policy implementation, while controlling for other possible influencing factors, enables an unbiased estimate of the policy effect [9]. For policy makers, evaluating the effectiveness of a particular policy by measuring the difference before and after the policy intervention facilitates subsequent policy modification and improvement.

The YREB Development Policy provides us with a quasi-natural experimental environment. We adopted 283 mainland Chinese cities as research samples, among which 107 cities in the YREB formed the experimental group, and the other 176 cities belonged to the control group. We applied the DID method to test the effects of YREB policy on industrial structure upgrading. We found that the YREB policy positively impacted industrial structure advancement but scarcely benefited industrial structure rationalization. Temporary and unsustainable impacts arise because the policy does not guarantee that different areas of the YREB will receive policy shocks equally. After a series of robustness tests, our results remain robust. Finally, an in-depth exploration of the effects of the implementation of the YREB Development Policy was conducted from three perspectives, namely, urban population size heterogeneity, regional heterogeneity, and industrial structure changes. The results showed that cities with larger populations are more conducive to industrial structure upgrading. Moreover, the effect of the industrial structure upgrading in the middle and lower reaches of the YREB is significantly better than that in the upper reaches. In addition, the YREB policy has accelerated the development of the tertiary industry in the YREB. This study is of great practical significance and can serve as a reference for other developing countries with similar economic development dynamics in China and globally.

The rest of this paper is organized as follows: The second section contains a brief literature review of relevant studies. The third section presents the research methods and data sources. The fourth section provides an analysis of the empirical results. The last section summarizes the findings and draws some research implications.

## 2. Literature Review

The industrial structure usually refers to the proportions of the three major industries in a country [10]. According to the international general classification standards, the industries in a country can be divided into three major categories. The primary industry usually includes the agriculture, forestry, animal husbandry and fishery industries; the secondary industry comprises industry and construction; all other activities belong to the tertiary industry. The upgrading of the industrial structure not only represents the core variable in understanding economic development differences among developed and developing countries but is also the essential requirement by which economically underdeveloped countries can accelerate their economic development [11]. Kuznets (1973) found that under modern economic growth conditions, the industrial structure is constantly adjusted as the economy develops [12]. The specific performance of the share of the primary industry in the total economic output continuously declines, the share of the secondary industry initially maintains growth and then gradually stabilizes, and the share of the tertiary industry in the total economic output continuously increases.

Previous studies have shown that the different dimensions of influencing factors can affect the upgrading of the industrial structure. Fisman (2003) and Antzpilatos (2011) found that financial development can promote technological progress in industries and there is a significant impact of financial development on industrial structural upgrading [13,14]. A few scholars have concluded that the urbanization level is closely related to the upgrading of the industrial structure. For example, Murakami (2015) empirically analysed the relationship between industrial structure changes and urbanization by using Japanese county data after World War II and found a two-way causality between the industrial structure and urbanization [15]. Urbanization may enable the regional economy to achieve a brief period of prosperity, but the positive effects of the constraints and limitations on the development of some industries are not sustainable [16]. Several scholars have explored the impact of information technology on the upgrading of industrial structure. For example, Dewan and Kraemer (2000) found that the better the integration of information technology with the local economic development model and industrial structure, the more obvious the promotion of the upgrading of the industrial structure [17]. However, Robert (2000) and Oulton (2002) argued that not all industries suffering from the impact of information technology are conducive to increasing productivity and upgrading the industrial structure [18,19]. Francois (1990) and Franke (2005) considered the active development of foreign trade as an effective path that can promote the participation of economies in the international division of labour and the transformation and upgrading of the industrial structure [20,21]. However, Schmitz (2004) concluded that actively boosting foreign trade does not have much impact on the upgrading of industrial structure [22]. Furthermore, some other research has indicated that social needs, social reforms, and policy changes may significantly impact the upgrading of the industrial structure.

Several studies have shown that public policy is the priority and a reliable tool for the structural upgrading of industries [23]. Cloete and Robb (2010) indicated that industrial policy can facilitate a shift in industrial structure towards high productivity activities, enabling South Africa to maximise its potential competitive advantage [24]. Craig (2015) also demonstrated that the increasing impact of industrial policy in optimising industrial structure [25]. Garcia and Coulter (2020) investigated the policy responses to industrial policy in four European countries and found significant differences and malleability of policies in terms of long-term impacts [26]. Vrolijk (2021) justified that inadequate policy implementation would hinder the industrial upgrading of underdeveloped economies and the findings implied that industrial policy needed to be improved and reconfigured [27]. Kenderdine (2017) stated that China’s industrial upgrading relies heavily on state-driven public policies to structure the economy [28].

In recent years, the Chinese government has actively proposed a series of public policies to promote the upgrading of industrial structure. Many scholars have noted the impact of environmental regulations on industrial structure. For example, Zhang et al., (2019) and Du et al., (2021) revealed the heterogeneous impacts of environmental regulations and environmental regulations on industrial structural upgrading at different economic development levels [29,30]. Some scholars have reported the impacts of industrial policies. For example, Chauffour and Maur (2011) found that free trade zones (FTZs) can fully utilize resources, remove barriers to the flow of production factors, goods and services, and lead to the expansion of nonprimary industries [31]. Other researchers have given attention to the impacts of infrastructure construction. For example, Kim (2000) found that the construction of high-speed rail networks accelerates the interregional flow of factor resources, thereby optimizing the regional resource allocation efficiency and increasing regional industrial output efficiency [32]. However, few empirical studies have systematically verified the potential impacts of national strategies on the upgrading of the industrial structure, especially in a major national strategic development area such as the YREB.

Although previous research into the effects of macro policies on the upgrading of China’s industrial structure has provided us with valuable information, some deficiencies still exist. First, most studies have focused on changes in explained variables within a certain period or after a policy has been issued, while the differences before and after policy implementations have not been accurately assessed [33]. Second, most studies have focused on the influence of a single factor on the explained variable while ignoring the influence of some unobservable factors; this kind of oversight leads to errors in the results [34]. Third, previous studies have given little attention to the effects of spatial heterogeneity, leading to less explicit and specific assessments [35]. Compared with other comprehensive evaluation methods, DID can not only effectively avoid the endogeneity problems caused by viewing policy as an explanatory variable but can also control the influence of unobservable variables that change over time [36]. The DID method also mitigates biases in missing variables and improves the accuracy of measurements. The YREB Development Policy aims to promote the balanced development of China’s regional economy and narrow the regional development gap. The policy has great potential to promote the industrial structure upgrading of the YREB. First, it is conducive to promoting the industry along the Yangtze River from a factor-driven industry to an innovation-driven industry and exerts a guiding role in the rational layout and orderly transfer of industries. Second, it is beneficial for leading the development of emerging industries and accelerating the transformation and upgrading of traditional industries. In addition, this policy also helps foster an international level of industrial clusters and enhance industrial competitiveness in the YREB. Based on this, this study regards the YREB Development Policy as a policy shock event and uses the DID method to identify the impact of the policy implementation on the upgrading of the industrial structure in the YREB. The purpose of this study is to fill this research gap in the literature and provide a reference for the improvement and adjustment of follow-up policies.

## 3. Methodology and Data

### 3.1. Data Source

Our data were mainly obtained from the China Urban Statistical Yearbook, the China Statistical Yearbook and provincial statistical yearbooks. This study adopts the data of 283 cities above the prefecture level (the Chinese mainland has 295 cities above the prefecture level) in Mainland China from 2009 to 2019 to investigate the impact of the YREB Development Policy on the upgrading of the industrial structure in the YREB. The specific geographical distribution of the YREB in China is shown in Figure 1. A portion of missing data were filled in by consulting official websites and using the average interpolation method. We excluded some prefecture-level cities, due to the serious lack of data. Considering that the YREB Development Policy was implemented in 2014, we set the starting time of the sample to 2009 to ensure the collection of a sufficient number of samples before the implementation of the policy. Finally, this study obtained a total of 3113 observation samples.

### 3.2. Econometric Model

This study used panel data from a total of 283 cities in Mainland China from 2009 to 2019 as a research sample. This paper defined 107 cities in the 11 provinces involved in the YREB as the experimental group. The remaining 176 cities not within the YREB are defined as the control group. We adopted the DID method to investigate whether the YREB Development Policy promoted the upgrading of the industrial structure in the YREB region. The setting method of the DID model, which compares the differences between the experimental group and the control group before and after the policy, is implemented by controlling for other factors. The DID model is set up as follows:(1)$${\mathrm{Upgrading}}_{\mathrm{it}}=\alpha_{0}+\alpha_{1}{\mathrm{treat}}_{i}\times {\mathrm{time}}_{t}+{\gamma X}_{\mathrm{it}}+\mu_{\mathrm{it}}+\pi_{\mathrm{it}}+\varepsilon_{\mathrm{it}}$$
where i represents the city and t represents the year. Upgrading_it_ is an explanatory variable that represents the industrial structure upgrading of the i-th city in the t-th year. This article measures industrial structure upgrading from two dimensions, namely, industrial structure rationalization and industrial structure advancement. The first layer of influence comes from the city, and the second layer of influence comes from the year. Therefore, this study sets two dummy variables. ${\mathrm{treat}}_{i}$ is the dummy variable set for the experimental group. When treat_i_ is 1, city i is one of 107 cities in the YREB. Similarly, if treat_i_ is 0, city i is not included among the 107 cities of the YREB. Time_t_ is a dummy variable representing the experimental period. When the time_t_ is 0, the policy has not yet been implemented. When the time_t_ is 1, the policy has been implemented. treat_i_ × time_t_ is the interaction term between treat_i_ and time_t_ and is also known as the treatment effect. When treat_i_ × time_t_ is 1, city i is one of the 107 cities in the YREB, and the policy has been implemented in city i in year t. When treat_i_ × time_t_ is 0, either city i is not among the cities in the YREB or the policy has not yet been implemented. Moreover, $X_{\mathrm{it}}$ is the control variable, $\mu_{\mathrm{it}}$ is the city fixed effect, and $\pi_{\mathrm{it}}$ is the year fixed effect. It is worth noting that $\alpha_{1}$ is the core estimation parameter and represents the net effect of the YREB Development Policy on the upgrading of the industrial structure. If $\alpha_{1}$ is greater than 0, it means that the implementation of the YREB Development Policy promoted industrial structure upgrading in the YREB. If it is less than 0, it means that the implementation of the YREB Development Policy had a reverse impact on the upgrading of the industrial structure in the YREB.

### 3.3. Variable Descriptions

#### 3.3.1. Dependent Variable

Our dependent variable is the upgrading of the industrial structure, and this variable can be divided into industrial structure advancement and industrial structure rationalization. The former refers to a transition between the economic growth mode and the development model [37]. According to Petty Clark’s law, with an increase in the per-capita national income, the centre of the national economy shifts from the primary industry to the secondary industry and then to the tertiary industry [38,39]. Therefore, this study adopted an industrial structure hierarchy coefficient (AIS) to measure industrial structure advancement. The calculation formula of this coefficient is as follows:(2)$${\mathrm{AIS}}_{\mathrm{it}}=\overset{3}{\underset{m=1}{\sum }}m\times y_{\mathrm{imt}},\;m=1,2,3$$
where y_imt_ stands for the proportion of the m industry in the regional GDP in city i in the t-th year. Equation (2) is the weighted sum of the proportion of the primary industry, the secondary industry and the tertiary industry, and the weights (3, 2, or 1) of the three industries are assigned according to the hierarchy level. When AIS is larger, the hierarchical coefficient of the industrial structure is larger, and the level of the industrial structure is higher.

Industrial structure rationalization refers to the process by which industrial synergy is improved through the adjustment of unreasonable industrial structures [40]. Using the Theil index to measure industrial structure rationalization can effectively retain its economic meaning; the specific calculation is shown in Equation (3):(3)$${\mathrm{Theil}}_{\mathrm{it}}=\overset{m}{\underset{i=1}{\sum }}\left(\frac{Y_{\mathrm{im}}}{Y_{i}}\right)\ln\left(\frac{Y_{\mathrm{im}}/Y_{i}}{L_{\mathrm{im}}/L_{i}}\right)$$
where $\frac{Y_{\mathrm{im}}}{Y_{I}}$ represents the output structure, that is, the proportion of the m-th industry in city i to the regional GDP, $\frac{L_{\mathrm{im}}}{L_{I}}\;$represents the employment structure, that is, the proportion of employed persons in the m-th industry in city i to all employed persons, and$\;\frac{L_{\mathrm{im}}}{L_{I}}\;$represents the employment structure, that is, the proportion of employment personnel in the m-th industry in city i to all employment personnel. Since the Theil Index is an inverse indicator, when the Theil Index is positive, the policy has inhibited the rationalization of the industrial structure, and vice versa.

#### 3.3.2. Independent Variable

The independent variable in this article is the product of the experimental group dummy variable and experimental period dummy variable (treat×time). This variable can be estimated based on the implementation time of the YREB Development Policy and whether the city under consideration is one of the 107 cities in the YREB. This is also the core explanatory variable in our article.

#### 3.3.3. Control Variables

Our control variables include the level of economic development, the process of urbanization, the level of opening up to the outside world, the level of infrastructure construction, the level of governmental financial support and the level of technological innovation; these factors involve economic, urban, open, infrastructure, innovation, government, and innovation dimensions, respectively. Descriptions of all control variables are listed in Table 1. The descriptive statistics of all variables are shown in Table 2.

#### 3.3.4. Correlation Test

This paper uses correlation tests to examine the correlation of the variables and to identify whether there is a problem of multicollinearity between the variables. We know that when a regression model has multicollinearity, the estimation standard errors may be inflated. This will lead to the problem of insignificant coefficient estimates occurring. After our tests we found that our regression model has a VIF value of 2.13, which is <10. Additionally, 1/VIF are higher than 0.2. Based on the above results we decided that there is no serious problem of multicollinearity between our variables. It means that our choice of variables is reasonable. The descriptive statistics of all variables are shown in Table 3.

## 4. Empirical Analysis

### 4.1. Parallel Trend Test

An important prerequisite for the use of the DID model in policy evaluations is that the experimental group and the control group must have common development trends before the policy is implemented. To verify this condition, we used the parallel trend tests proposed by Beck and Jacobson. As shown in Figure 2, before 2014, the parameter estimation coefficient of the advancement of the industrial structure in the YREB fluctuated around approximately 0, and the confidence interval of the coefficient included 0. This indicates that before the YREB Development Policy was implemented, there was no significant difference between the experimental group and the control group. Therefore, the advancement of the industrial structure passed the parallel trend test. In the three years following the implementation of the policy, AIS showed a clear “U-shaped” development trend, indicating that the implementation of the policy did, in fact, impact the advancement of the industrial structure. Only the year when the policy was implemented and the following two years showed a significant negative impact; subsequently, the policy promoted the industrial structure upgrading process. Then, it returned to the nonsignificant state, indicating that the impact of the YREB Development Policy on industrial structure advancement was temporary and unsustainable.

It can be seen from Figure 3 that before the policy was implemented, the estimated parameter coefficients of the rationalization of the industrial structure all fluctuated around approximately 0, and the confidence intervals of the coefficients all contained 0. This indicates that there was no significant difference between the experimental group and the control group before the implementation of the policy, and the industrial structure rationalization thus passed the parallel trend test. However, following the implementation of the policy, many years of parameter estimates still fluctuated around approximately 0, and the confidence intervals of the coefficients also contained 0. This result suggests that the YREB Development Policy did not play a role in promoting industrial structure rationalization after its implementation. Only in the third year following the implementation of the policy did the policy have a significant inhibitory impact.

### 4.2. Base Regression Analysis

Columns (1) and (2) in Table 3 show the base regression results representing the impact of the YREB Development Policy on industrial structure advancement. In column (1), the coefficient of Treat × Time representing the AIS treatment effect was 0.027, which was significant at the 5% level. This means that after the implementation of the YREB Development Policy, the industrial structure advancement in the YREB increased by 2.7%. After adding a series of control variables, the estimated value of the treatment effect coefficient was reduced to 0.017, which was still significant at the 5% level. This result confirmed that the YREB Development Policy did indeed significantly promote industrial structure advancement.

Columns (3) and (4) in Table 3 show the base regression results of the impact of the YREB Development Policy on industrial structure rationalization. The results indicate that before the control variables were added, the coefficient of Treat × Time representing the Theil treatment effect in column (3) was −0.007; this value was not statistically significant. After a series of control variables were added, the Treat × Time coefficient in column (4) was still not statistically significant. This result confirmed that the YREB Development Policy did not significantly affect industrial structure rationalization in the YREB.

In addition, columns (2) and (4) in Table 4 report the impacts of the control variables on the upgrading of the industrial structure. In column (2), except for the level of technological innovation, the other six control indicators all positively impacted industrial structure advancement. In column (4), four control variables significantly affected industrial structure rationalization, namely, the level of economic development, the process of urbanization, the level of opening up, and the level of infrastructure construction.

### 4.3. Robustness Test

#### 4.3.1. Placebo Test

A series of robustness tests were carried out to ensure the robustness of the results. First, a placebo test was performed by adjusting the time window to construct a dummy policy implementation year. The implementation year of the policy was assumed to be 2008, and data from 2005 to 2011 were selected to examine the impact of the policy on the upgrading of the industrial structure in the YREB. Table 5 shows the results of this test obtained after adjusting the time window. Column (1) shows that before the control variables were added, the AIS coefficient was −0.006, which was not statistically significant. After adding the control variables, the results listed in column (2) were still not statistically significant, contrary to the results listed in Table 3. In addition, the test results of the Theil index showed opposite results. The above results confirm that the study passed the placebo test.

#### 4.3.2. Adjusted Sample Size

Table 6 shows the test results obtained after adjusting the sample size. Since non-random problems may exist in central cities, the “peripheral cities” method was adopted: the samples of all provincial capitals and four municipalities were eliminated, and the DID test was performed again. We found that after removing these cities, the impacts of the YREB Development Policy on the AIS and Theil index were consistent with the regression results listed in Table 3. This indicates that the inclusion of provincial capitals and municipalities did not cause measurement errors in the test results. Therefore, the results of this test confirm the robustness of our findings.

### 4.4. Heterogeneity Analysis

#### 4.4.1. Analysis of Population Heterogeneity

Due to China’s vast territory and large population, great developmental differences exist among regions. Therefore, in this study, we divided the research samples according to the size of the urban population. If the urban population was greater than 5 million, this article defined the urban area as a large city. Cities with populations under 5 million were considered smaller cities. After controlling the control variables, city fixed effect and year fixed effect tests were carried out sequentially, and the regression results are shown in Table 7. The results show that in cities with large urban populations, the YREB Development Policy had a positive impact (coefficient = 0.023, *p* < 0.05) on industrial structure advancement. For industrial structure rationalization, the impact of the YREB Development Policy was nonsignificant. For cities with small urban populations, the interaction coefficients of industrial structure advancement and rationalization were both nonsignificant. These results indicate that the implementation of the YREB Development Policy had a more significantly positive impact on industrial structure advancement in cities with large populations than in cities with small populations.

#### 4.4.2. Analysis of Regional Heterogeneity

Traditionally, people divide the YREB into upper, middle, and lower reaches based on geographical and economic factors. The upper reaches of the Yangtze River include the four provinces of Yunnan, Guizhou, Sichuan, and Chongqing. The middle reaches include Hunan, Hubei, and Jiangxi Provinces. The lower reaches involve Shanghai, Jiangsu, Zhejiang, and Anhui Provinces. In our article, 107 cities in 11 provinces within the YREB were included as the research objects, and research was carried out according to the heterogeneity among these cities.

Table 8 shows the impact of the YREB Development Policy on the upgrading of the industrial structure across the upper, middle and lower reaches of the YREB. In the upper reaches of the YREB, judging from the coefficient of the interaction term, the implementation of the policy did not have a significant impact (AIS coefficient of −0.005, *p* > 0.1; Theil coefficient of 0.028, *p* > 0.1). In the middle reaches of the YREB, the policy significantly positively impacted industrial structure advancement (AIS coefficient of 0.025, *p* < 0.1) but had a nonsignificant impact on industrial structure rationalization (Theil coefficient of −0.009, *p* > 0.1). In the lower reaches of the YREB, the policy significantly positively impacted industrial structure advancement (AIS coefficient of 0.027, *p* < 0.05) but had a nonsignificant impact on industrial structure rationalization (Theil coefficient of 0.009, *p* > 0.1). It can be seen from these results that the policy more obviously impacted economically active areas and even had restrictions on the transformation and upgrading of the upper and middle reaches.

#### 4.4.3. Analysis of Industrial Structure Change

The above results indicate that the implementation of the YREB Development Policy more significantly impacted industrial structure advancement than industrial structure rationalization. Therefore, this study considered the proportions of the primary, secondary, and tertiary industries in the regional GDP as proxy variables representing industrial structure changes to explore the changes induced by the policy on the industrial structure of the YREB. Columns (1), (2), and (3) in Table 9 successively represent the impacts of the YREB Development Policy on the primary, secondary and tertiary industries in the YREB. In columns (1) and (2), the proportions of the primary and secondary industries decrease following the implementation of the policy, but these decreases are not significant. In contrast, the YREB Development Policy promoted the growth of the tertiary industry (coefficient = 0.013, *p* < 0.05). Consequently, this result implies that the development focus of the industrial structure tends to be service-oriented, and this focus benefits the development of the advanced industrial structure.

As clarified above, the YREB Development Policy promoted the development of the tertiary industry in cities in the lower reaches of the YREB and significantly reduced the primary industry proportion in the middle reaches and the secondary industry proportion in the lower reaches. To further determine the roles these control variables play in the upgrading of the industrial structure among different regions, we conducted a deeper analysis of the regional industrial structure changes (see Table 10).

Economy. In the middle and upper reaches of the YREB, the development speed of the secondary industry is almost two to three times that of the tertiary industry. However, in the lower reaches of the YREB where the economic strength is robust, the development of the secondary and tertiary industries does not differ extensively and is relatively balanced.Urban. Accelerating the process of urbanization in the YREB can restrict the development of the primary and secondary industries and can play a sustained and significant role in promoting the tertiary industry. Under this policy shock, the impacts of urbanization on the middle and upper reaches of the YREB were more pronounced than that on the lower reaches. The impacts on the secondary and tertiary industries were obviously more significant than that on the primary industry.Opening up. The results show that the impacts of the level of opening up on the three industries in the upper and middle reaches of the YREB were relatively consistent. An increase in the level of opening up significantly reduced the primary and secondary industry proportions and promoted the development of the tertiary industry. Moreover, under the current policy shock, the development of the tertiary industry in the upper reaches of the YREB is better than that in the middle reaches, while the policy has no significant impact on the lower reaches.Infrastructure. As shown in Table 9, the higher the level of infrastructure construction is, the more constrained the primary and secondary industries are. The level of infrastructure construction can promote the vigorous development of the tertiary industry. For China, accelerating the infrastructure construction level means promoting the trans-regional flow of resources, and this promotion helps accelerate the regional integration process and is conducive to driving the rapid development of the regional economy.Innovation. The empirical results indicate that the implementation of the YREB Development Policy significantly increased the number of patent applications in the primary industry of the YREB and thus improved the technological innovation level of the primary industry. The policy also significantly reduced the enthusiasm of patent applications in the secondary industry.Governmental support. In the process of promoting the transformation of the industrial structure, the government has been vigorously developing the primary and tertiary industries. The YREB Development Policy has played an active role in guiding regional modern agriculture and developing the modern service industry. However, the government has significantly reduced its funding support for the secondary industry, which contains a large number of highly energy-consuming industries with underdeveloped production capacities.Informatization. One of the intentions of the implementation of the YREB Development Policy was to make full use of the new generation of information technology to transform and upgrade traditional industries and cultivate emerging industries. As shown in Table 9, the positive impact of improved informatization on the industrial structure is embodied in the tertiary industry. Informatization also greatly weakened the primary industry and secondary industry proportions.

## 5. Conclusions and Research Implications

### 5.1. Conclusions

Currently, the transformation of the economic development mode has become a top priority in China, and this transformation is closely tied to the upgrading of the industrial structure. Based on panel data characterizing 283 cities in Mainland China from 2009 to 2019, this article explored the changes in the industrial structure of the YREB from the perspectives of industrial structure rationalization and industrial structure advancement by using the DID method. To further explore the implementation effects of the YREB Development Policy, an in-depth exploration of the policy implementation effects was conducted from three perspectives, namely, the urban population size heterogeneity, regional heterogeneity, and industrial structure changes.

First, this study revealed that the YREB Development Policy significantly promoted the industrial structure advancement of the YREB, but its impact on industrial structure rationalization has been nonsignificant. Our finding implies a hysteresis effect exerted by the YREB Development Policy on the upgrading of the industrial structure. This finding was consistent with the results reported by Zheng et al., (2021), who found that pilot programmes in low-carbon cities positively impacted industrial structure supererogation but had little effect on industrial structure rationalization [41]. At present, the development of China’s tertiary industry basically goes hand in hand with the development of the secondary industry, and the tertiary industry is even developing faster than the secondary industry [42]. As the tertiary industry gradually replaces the dominant position held by the secondary industry, the succession characteristics of the industrial structure are gradually emerging [43]. After controlling some latent influencing factors and conducting a series of robustness tests, our results were still reliable.

Second, this study found that the YREB Development Policy imposed a short-lived and unsustainable impact on industrial structure advancement, presenting an obvious U-shaped development trend. On the one hand, the macroeconomic situation changed due to the time lag, causing an opposite effect to be observed following the policy implementation in that year [44]. It was not until the third year following the implementation of the YREB Development Policy that a significant positive impact on industrial structure advancement was observed. On the other hand, depending on the policy itself, certain lag periods occur regarding the manifestation of policy effects [45]. Our results support Garcia and Coulter’s study that policy outcomes can be either short-term politicised remedies or enduring policy changes, and that policy responses varied significantly in terms of coherence and long-term impact [26]. Previous research has confirmed that if policy makers are only concerned about recent policy benefits and that this shortsightedness provokes inefficient public policy [46,47]. In this study, since the policy was fully enacted in the YREB region, the policy ignores differences in the responses of different regions to policy shocks, resulting in the measured temporary and unsustainable effects of the policy.

Third, this study confirmed that cities with larger populations in the YREB were more conducive to industrial structure upgrading. This finding is consistent with previous studies that population plays a significant role in the integral upgrading of the tertiary industry and that the scarcity of human resources will cause the integral upgrading lags far behind [48,49]. Additionally, our population-scale heterogeneous analysis revealed that the development speed in the middle and lower reaches of the YREB was significantly higher than that of the upper reaches; moreover, the effects of industrial structure advancement in the middle and lower reaches were almost five times that in the upper reaches. Our findings are in line with the conclusion of Jin et al., (2018), who confirmed that the YREB development efficiency conforms to a “bar-like” distribution across the whole area, gradually decreasing from the east to the west [50]. Furthermore, the level of economic development, the process of urbanization, the level of opening up to the outside world, the level of infrastructure construction, governmental financial support and the level of technological innovation all positively impacted the development of the tertiary industry in different regions in the YREB. It means that the YREB Development Policy has clearly promoted the priority development of the tertiary industry and accelerated the service-oriented economic development trend in the YREB region.

### 5.2. Research Implications

Our potential contributions to the literature are reflected in the following aspects. First, this study organically links the YREB Development Policy with the development of the regional industrial structure and accurately examines the effects of this policy from the perspectives of industrial structure rationalization and industrial structure advancement. The findings fill the research gaps regarding the impacts of national macro policies on the upgrading of the regional industrial structure and contribute to the further improvement of follow-up policies. Second, this study further compared the differentiated impacts of the YREB Development Policy among regions from the perspectives of the urban population scale and regional heterogeneity. We found that the YREB Development Policy more significantly impacted cities with larger urban populations. Additionally, we revealed the generally positive impacts of a series of control variables on the development of the tertiary industry across the upper, middle and lower reaches of the YREB. These findings provide an empirical reference with which more effective development policies with local characteristics can be formulated in the future. Third, this study proposes a more scientific policy-evaluation method to evaluate the effectiveness of the YREB Development Policy. Most previous studies typically ignored the impacts of some unobservable factors and tried to eliminate endogeneity problems simply by including measurements themselves [51]; these methodologies may have caused one-sidedness in the research results. This study considered the YREB Development Policy as an exogenous shock and applied the DID method to assess the impact of the policy on the upgrading of the industrial structure, thus maximally avoiding measurement errors [52,53].

Inspired by the research findings, several practical suggestions can be proposed. First, policymakers should realize that administrative divisions are an important obstacle to high-quality development. Local governments should reject low-level and inefficient redundant construction and simultaneously formulate new policies that are in line with local characteristics based on the development advantages and economic foundation of the corresponding cities. Second, to achieve the high-quality development of the regional economy, local governments should consider the importance of their technological innovation ability and information construction, unswervingly follow the path of innovation-driven optimization and the upgrading of the industrial structure, and continuously improve the local informatization level. Third, government departments should give policy preference to the underdeveloped areas in the upper and middle reaches of the YREB when making strategic arrangements. Departments can guide the upgrading of the industrial structure in the middle and upper reaches of underdeveloped areas by relying on the advantages of capital, technology and talent and can continuously narrow the development gap between the upper, middle and lower reaches of the YREB to promote the overall balanced development of the YREB.

## Figures and Tables

**Figure 1 ijerph-19-05042-f001:**
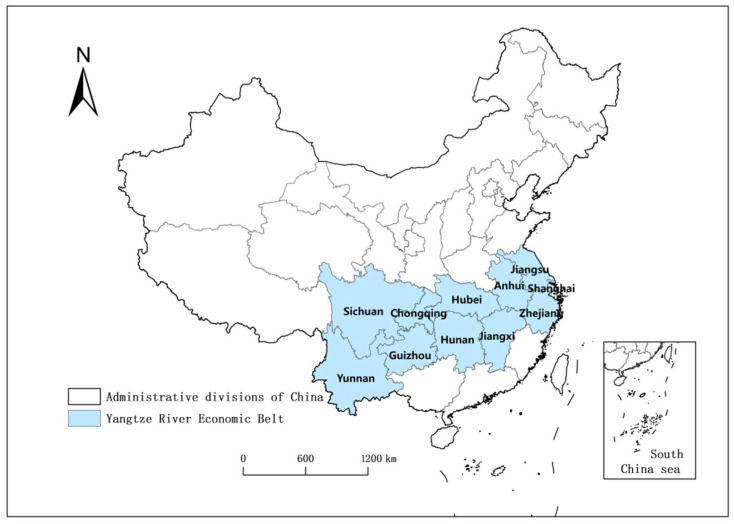
Geographic distribution of the YREB in China.

**Figure 2 ijerph-19-05042-f002:**
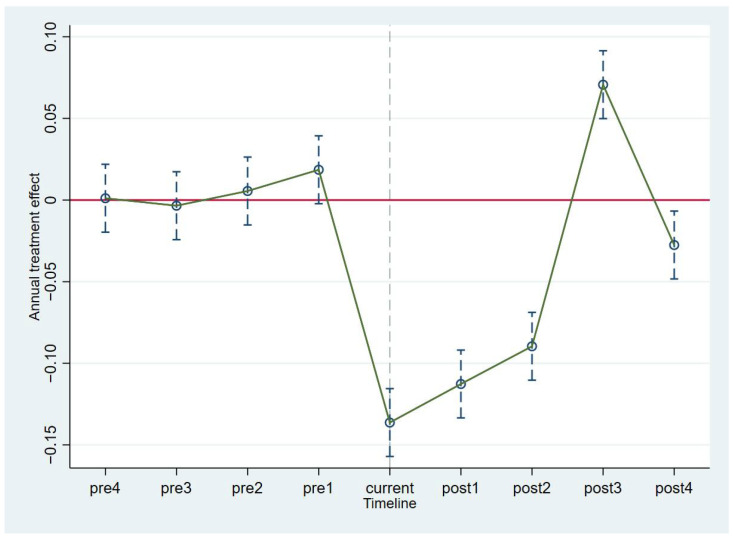
Parallel trend test for AIS.

**Figure 3 ijerph-19-05042-f003:**
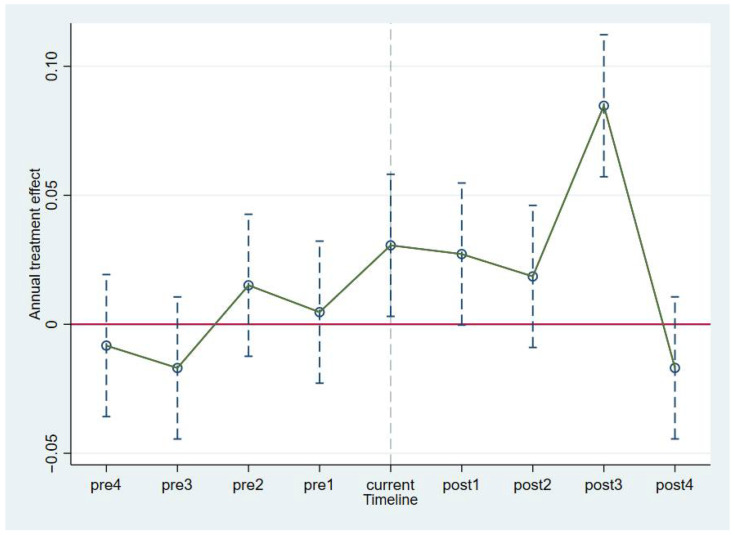
Parallel trend test results on the Theil index.

**Table 1 ijerph-19-05042-t001:** Descriptions of control variables.

Dimension	Indicator	Description
Economy	The level of economic development	GDP per capita
Urban	The process of urbanization	The proportion of the urban population to the total regional population
Open	The level of opening up	The proportion of foreign capital to the total regional GDP
Infrastructure	The level of infrastructure construction	The ratio of the built-up area to the total regional population
Innovation	The level of technological innovation	Patents per capita
Government	The level of governmental support	The proportion of financial expenditure to the total regional GDP
Information	The level of informatization	The ratio of post and telecommunications per capita to the GDP per capita

**Table 2 ijerph-19-05042-t002:** Descriptive statistics.

Variable	N	Mean	Sd	Min	Max
AIS	3113	2.282	0.154	1.831	2.986
Theil	3113	0.274	0.211	−0.080	1.720
Treat × Time	3113	0.202	0.402	0.000	1.000
Economy	3113	10.58	0.729	8.391	13.19
Urban	3113	0.253	0.280	0.000	3.986
Open	3113	0.018	0.018	0.000	0.193
Infrastructure	3113	3.221	0.858	1.236	9.558
Innovation	3113	0.001	0.002	0.000	0.026
Government	3113	0.230	0.119	0.0188	1.515
Information	3113	0.025	0.019	0.000	0.274

**Table 3 ijerph-19-05042-t003:** Correlation test.

Variable	VIF	1/VIF
Economy	3.87	0.26
Urban	2.37	0.42
Open	1.12	0.89
Infrastructure	3.15	0.32
Innovation	1.86	0.54
Government	1.43	0.70
Information	1.10	0.91
Mean VIF	2.13	

**Table 4 ijerph-19-05042-t004:** Impact of the YREB Development Policy on the upgrading of the industrial structure.

	(1)	(2)	(3)	(4)
Variable	AIS	AIS	Theil	Theil
Treat × Time	0.027 **	0.017 **	−0.007	0.011
	(2.48)	(2.07)	(−0.46)	(0.88)
Economy		0.100 ***		−0.118 ***
		(15.66)		(−12.72)
Urban		0.068 ***		−0.035 **
		(3.83)		(−2.35)
Open		0.680 ***		−0.979 ***
		(5.29)		(−5.49)
Infrastructure		0.016 ***		−0.067 ***
		(3.15)		(−10.41)
Innovation		0.199		17.085 ***
		(0.12)		(8.17)
Government		0.083 ***		−0.014
		(3.69)		(−0.38)
Information		1.165 ***		−0.315
		(8.16)		(−1.58)
Constant	2.228 ***	1.060 ***	0.252 ***	1.717 ***
	(474.07)	(17.17)	(35.50)	(18.61)
Observations	3113	3113	3113	3113
R-squared	0.113	0.483	0.018	0.375

Note: *** *p* < 0.01 and ** *p* < 0.05.

**Table 5 ijerph-19-05042-t005:** Test results obtained after adjusting the time window.

	(1)	(2)	(3)	(4)
Outcome Var.	AIS	AIS	Theil	Theil
Before				
Control	2.195	1.233	0.251	1.568
Treated	2.193	1.262	0.282	1.547
Diff (T-C)	−0.001	0.029	0.031 **	−0.021 *
After				
Control	2.220	1.213	0.239	1.614
Treated	2.212	1.228	0.304	1.633
Diff (T-C)	−0.008	0.015	0.066 ***	0.019
Diff-in-Diff	−0.006	−0.014	0.035 *	0.039 **

Note: *** *p* < 0.01, ** *p* < 0.05, and * *p* < 0.1.

**Table 6 ijerph-19-05042-t006:** Test results obtained after adjusting the sample size.

	(1)	(2)	(3)	(4)
Outcome Var.	AIS	AIS	Theil	Theil
Before				
Control	2.199	1.214	0.275	1.782
Treated	2.198	1.226	0.330	1.803
Diff (T-C)	−0.001	0.012 **	0.055 ***	0.021 ***
After				
Control	2.293	1.266	0.027	1.833
Treated	2.320	1.295	0.322	1.867
Diff (T-C)	0.027 ***	0.029 ***	0.046 ***	0.034 ***
Diff-in-Diff	0.028 ***	0.017 **	−0.009	0.012

Note: *** *p* < 0.01 and ** *p* < 0.05.

**Table 7 ijerph-19-05042-t007:** Test results of the heterogeneity of the urban population size.

	Population ≥ 500		Population < 500	
Variable	(1)	(2)	(3)	(4)
	AIS	Theil	AIS	Theil
Treat × Time	0.023 **	0.021	0.012	−0.001
	(2.23)	(1.22)	(1.22)	(−0.06)
Control	Yes	Yes	Yes	Yes
City fixed effect	Yes	Yes	Yes	Yes
Year fixed effect	Yes	Yes	Yes	Yes
Constant	1.075 ***	1.692 ***	1.150 ***	1.579 ***
	(16.03)	(17.33)	(17.22)	(16.27)
Observations	2497	2497	2552	2552
R-squared	0.511	0.394	0.452	0.356

Note: *** *p* < 0.01 and ** *p* < 0.05.

**Table 8 ijerph-19-05042-t008:** Test results obtained when considering urban heterogeneity.

	The Upper Reaches		The Middle Reaches		The Lower Reaches	
Variable	(1)	(2)	(3)	(4)	(5)	(6)
	AIS	Theil	AIS	Theil	AIS	Theil
Treat × Time	−0.005	0.028	0.025 *	−0.009	0.027 **	0.009
	(−0.40)	(1.28)	(1.88)	(−0.47)	(2.57)	(0.52)
Control	Yes	Yes	Yes	Yes	Yes	Yes
City fixed effect	Yes	Yes	Yes	Yes	Yes	Yes
Year fixed effect	Yes	Yes	Yes	Yes	Yes	Yes
Constant	1.202 ***	1.606 ***	1.195 ***	1.519 ***	1.069 ***	1.612 ***
	(16.89)	(16.18)	(17.12)	(15.68)	(16.10)	(15.87)
Observations	2265	2265	2332	2332	2386	2386
R-squared	0.484	0.423	0.450	0.351	0.510	0.366

Note: *** *p* < 0.01, ** *p* < 0.05, and * *p* < 0.1.

**Table 9 ijerph-19-05042-t009:** Test results obtained when considering industrial structure changes.

Variable	(1)	(2)	(3)
	PI	SI	TI
Treat × Time	−0.004	−0.009	0.013 **
	(−1.14)	(−1.28)	(1.99)
Control	Yes	Yes	Yes
City fixed effect	Yes	Yes	Yes
Year fixed effect	Yes	Yes	Yes
Constant	0.873 ***	0.194 ***	−0.071
	(29.46)	(3.26)	(−1.34)
Observations	3113	3113	3113
R-squared	0.585	0.297	0.328

Note: *** *p* < 0.01 and ** *p* < 0.05.

**Table 10 ijerph-19-05042-t010:** Test results obtained when considering regional industrial structure changes.

Variable	The Upper Reaches			The Middle Reaches			THE Lower Reaches		
	(1)	(2)	(3)	(4)	(5)	(6)	(7)	(8)	(9)
	PI	SI	TI	PI	SI	TI	PI	SI	TI
Treat × Time	0.006	−0.008	0.003	−0.009 *	−0.007	0.016	−0.006	−0.016 *	0.023 **
	(1.10)	(−0.74)	(0.25)	(−1.88)	(−0.64)	(1.46)	(−1.50)	(−1.69)	(2.43)
Economy	−0.069 ***	0.053 ***	0.017 ***	−0.069 ***	0.051 ***	0.018 ***	−0.071 ***	0.040 ***	0.031 ***
	(−18.72)	(8.49)	(2.91)	(−18.77)	(8.36)	(3.28)	(−19.92)	(6.16)	(5.46)
Urban	−0.011 *	−0.062 ***	0.073 ***	−0.009	−0.066 ***	0.075 ***	−0.008 *	−0.061 ***	0.070 ***
	(−1.71)	(−4.57)	(4.27)	(−1.47)	(−4.75)	(4.36)	(−1.72)	(−5.22)	(4.92)
Open	−0.184 ***	−0.474 ***	0.674 ***	−0.270 ***	−0.234 *	0.520 ***	−0.100 **	0.011	0.105
	(−3.39)	(−3.34)	(4.95)	(−4.99)	(−1.78)	(4.02)	(−2.06)	(0.09)	(0.88)
Infrastructure	−0.006 **	−0.008 *	0.014 ***	−0.005 *	−0.005	0.010 **	−0.005 *	−0.002	0.007 *
	(−2.00)	(−1.93)	(3.22)	(−1.87)	(−1.12)	(2.31)	(−1.73)	(−0.59)	(1.72)
Innovation	3.433 ***	−4.610 ***	1.236	3.463 ***	−4.529 ***	1.129	3.586 ***	−4.861 ***	1.314
	(6.27)	(−2.76)	(0.76)	(6.29)	(−2.74)	(0.72)	(7.99)	(−3.22)	(0.92)
Government	0.119 ***	−0.308 ***	0.191 ***	0.117 ***	−0.319 ***	0.202 ***	0.121 ***	−0.306 ***	0.185 ***
	(8.08)	(−15.60)	(11.32)	(7.24)	(−15.33)	(11.72)	(6.92)	(−10.57)	(8.87)
Information	−0.205 ***	−0.610 ***	0.805 ***	−0.234 ***	−0.711 ***	0.934 ***	−0.240 ***	−0.724 ***	0.953 ***
	(−3.50)	(−4.73)	(6.79)	(−4.15)	(−5.05)	(7.38)	(−4.24)	(−5.11)	(7.54)
Constant	0.851 ***	0.092	0.052	0.855 ***	0.097	0.044	0.867 ***	0.196 ***	−0.067
	(24.81)	(1.50)	(0.92)	(24.69)	(1.60)	(0.79)	(25.37)	(2.93)	(−1.16)
Observations	2662	2662	2662	2332	2332	2332	2387	2387	2387
R-squared	0.539	0.345	0.363	0.534	0.342	0.333	0.571	0.329	0.362

Note: *** *p* < 0.01, ** *p* < 0.05, and * *p* < 0.1.

## Data Availability

The datasets used or analysed during the current study are available from the corresponding author on reasonable request.

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
