# Peer review of "Examining the Impact of Regional Development Policy on Industrial Structure Upgrading: Quasi-Experimental Evidence from China"

_ijerph, 2022, doi:10.3390/ijerph19095042_

Round 1
Reviewer 2 Report
This is a very interesting paper that uses robust econometric methods to identify effects of the YREB development policy in China on regional structural change. The sections on methodology, empirical results and discussion and research implications are fine.
I would recommend that the authors sharpen a bit the introduction, by omitting some references that are not strongly linked to the topic of the paper and by providing a bit more explanation of the reliable econometric approach that is used in the paper. Also, the introduction should explain how the paper is structured and provide a brief summary of the main findings.
Another suggestion is that the literature review should also discuss some studies that have examined policy making on (regional) restructuring or upgrading in western economies, so that more readers will be able to appreciate the wider importance and relevance of the findings in the paper. Some links with the broader research on (regional) policy making and its effects will achieve this.
